# TAMER: A Tri-Modal Contrastive Alignment and Multi-Scale Embedding Refinement Framework for Zero-Shot ECG Diagnosis

## Abstract

Cardiovascular disease (CVD) diagnosis relies heavily on electrocardiograms (ECGs). However, most existing self-supervised uni-modal methods suffer from limited representational capacity, while multi-modal frameworks are hindered by coarse-grained semantic alignment across modalities, thus restricting their generalizability in clinical settings. To address these limitations, we propose TAMER, a **T**ri-modal contrastive **A**lignment and **M**ulti-scale **E**mbedding **R**efinement framework that jointly models ECG recordings, spectrograms, and diagnostic reports. TAMER is composed of three key components: First, the tri-modal feature encoding and projection (TFEP) module employs modality-specific encoders to extract global and local features from ECG recordings, spectrograms, and diagnostic reports, and projects them into latent spaces. Then, the global-local temporal-spectral alignment (GLTSA) module captures complementary rhythm- and wave-level characteristics via contrastive alignment and attentive interaction between temporal and spectral modalities. Finally, the report-aware alignment and refinement (RAAR) module performs diagnostic-level alignment and wave-level refinement with clinical reports, enabling semantic enrichment of ECG representations. Extensive experiments on three public ECG datasets demonstrate that TAMER achieves state-of-the-art zero-shot classification performance (AUC: 81.2%) and strong cross-domain generalization (AUC: 83.1%), outperforming existing uni-modal and multi-modal baselines methods. The source code is available at https://anonymous.4open.science/r/TAMER-FB58.

## 1 Introduction

Early detection of cardiovascular diseases (CVDs) is critical for improving patient outcomes and reducing healthcare costs Tripathi et al. (2022); Elvas et al. (2025). Subtle irregularities in ECG recordings can indicate early signs of arrhythmia, ischemia, and other cardiac abnormalities, often before the onset of severe symptoms Acharya et al. (2016); Hong et al. (2020); Yagi et al. (2024). Consequently, ECG serves as a vital tool for identifying at-risk individuals, enabling timely interventions that help prevent disease progression and reduce mortality.

In recent years, the increasing availability of clinical data and advances in deep learning have led to significant progress in automated ECG diagnostic models Ribeiro et al. (2020); Huang et al. (2022); Liu et al. (2023); Al-Zaiti et al. (2023); Ameen et al. (2024). However, several challenges continue to hinder their widespread clinical adoption. First, the scarcity of labeled data for specific or rare clinical conditions poses a major obstacle, making it difficult to train reliable models that generalize well across diverse patient populations. Second, uni-modal ECG signals, which primarily reflect electrical activity, not only fail to capture the complex structural and functional abnormalities associated with cardiovascular disease, but also suffer from inherent noise and variability that hinder effective multi-modal fusion Zhang et al. (2023); Tripathi et al. (2022); Ameen et al. (2024).

To mitigate the scarcity of annotated data, self-supervised learning (SSL) has emerged as a powerful paradigm for ECG representation learning Wang et al. (2023); Zhang et al. (2022). Existing ECG SSL approaches generally fall into two categories: contrastive learning Mehari & Strodthoff (2022); Li et al. (2022); Oh et al. (2022) and generative learning Zhang et al. (2023); Na et al. (2024).

Contrastive methods learn discriminative representations by constructing positive and negative pairs in the embedding space, whereas generative methods rely on masked reconstruction tasks to model latent structural patterns. However, these methods predominantly focus on uni-modal time-series data, limiting their capacity to capture complex pathological features.

To effectively leverage other modalities, recent research has focused on multi-modal ECG modeling (Zhang et al., 2023; Lalam et al., 2023). One promising approach involves extracting time-frequency joint representations Bui et al. (2024); Yang et al. (2024), which improve sensitivity to local perturbations and non-stationary rhythms. Another emerging trend incorporates clinical diagnostic reports Liu et al. (2024); PHAM et al. (2024), providing high-level semantic supervision. However, several key challenges remain unresolved: (1) Temporal and spectral modalities emphasize distinct feature types and are subject to modality-specific noise Singh & Krishnan (2023), leading to semantic misalignment and fusion instability. (2) Most ECG-report alignment methods focus primarily on global coarse matching, neglecting local correspondences between waveform anomalies and diagnostic phrases Liu et al. (2024); PHAM et al. (2024), which limits the detection of subtle abnormalities.

To address these challenges, we propose TAMER, a tri-modal contrastive alignment and multi-scale embedding refinement framework for zero-shot ECG diagnosis. TAMER is composed of three key components: (1) The tri-modal feature encoding and projection (TFEP) module employs modality-specific encoders and projections to extract global and local features from ECG recordings, spectrograms, and clinical reports, projecting each into latent spaces. (2) The global-local temporal-spectral alignment (GLTSA) module performs rhythm-level contrastive alignment and wave-level attentive interaction between temporal and spectral ECG features, producing a unified ECG representation that captures multi-scale diagnostic patterns. (3) The report-aware alignment and refinement (RAAR) module integrates report-anchored diagnostic-level alignment and report-guided wave-level refinement to enable semantic awareness, yielding robust ECG representations.The main contributions are summarized as follows:

- We propose a tri-modal self-supervised ECG framework that jointly models ECG recordings, spectrograms, and clinical reports, extracting complementary diagnostic information from underexplored modalities.

- We introduce the GLTSA and RAAR modules to improve tri-modal ECG representations by enforcing temporal-spectral consistency and enabling cross-modal, global-local semantic alignment between ECG signals and clinical reports.

- Extensive experiments conducted on three public ECG datasets demonstrate that TAMER outperforms state-of-the-art methods in zero-shot classification and cross-domain generalization, highlighting its strong transferability and clinical relevance.

## 2 RELATED WORK

### 2.1 SELF-SUPERVISED LEARNING IN ECG ANALYSIS

In recent years, SSL, broadly categorized into contrastive and generative methods, has made notable progress in intelligent ECG diagnosis. For contrastive learning, Mehari & Strodthoff (2022) adapted classical visual contrastive techniques to ECG data, demonstrating their feasibility and effectiveness in modeling medical time-series signals. Meanwhile, Wang et al. (2023) proposed ASTCL, which enhances the robustness and spatiotemporal representation of ECG signals via adversarial contrastive learning. However, contrastive methods often rely on augmentations that can introduce non-physiological features, reducing sensitivity to critical patterns. In contrast, generative approaches such as ST-MEM Na et al. (2024) and MAFE Zhang et al. (2022) utilize Vision Transformers and masking strategies to reconstruct occluded segments, capturing local morphological patterns and temporal dependencies. Moreover, CRT Zhang et al. (2023) and MassMIB Yang et al. (2024) performed cross-domain reconstruction of time- and frequency-domain representations, exploiting their complementary characteristics to enhance robustness of ECG representations. However, these methods typically demand significant computational resources and large-scale datasets.

Overall, current ECG SSL methods limited in capturing complex or subtle abnormal patterns due to underutilization of diverse information inherent in clinical data, highlighting the need for multi-modal frameworks integrating time-domain signals, frequency-domain features, and clinical context.

## 2.2 MULTI-MODAL LEARNING FOR ECG ANALYSIS

The multi-modal nature of ECG data is increasingly recognized as a key factor in enhancing diagnostic performance. Integrating ECG signals, spectrograms, and clinical reports yields more comprehensive representations, motivating current multi-modal methods. On the one hand, frequency-domain features complement time-domain signals by capturing rhythmic and instantaneous frequency variations (Yang & Hong, 2022; Duan et al., 2024; Zhou et al., 2024). This has motivated ECG-specific time-frequency model methods such as CRT Zhang et al. (2023) and MassMIB Yang et al. (2024), which employ masked reconstruction across time and frequency views to capture global context and enhance cross-view robustness. On the other hand, inspired by CLIP Radford et al. (2021), vision-language contrastive learning has been widely adopted in medical image analysis, where clinical reports are increasingly regarded as a key modality for providing high-level diagnostic supervision (Wang et al., 2022; Cheng et al., 2023). More recently, MERL Liu et al. (2024) extended this paradigm to ECG data, introducing a multimodal contrastive learning framework that aligns ECG signals and clinical reports in a shared embedding space for zero-shot diagnosis.

In clinical practice, diagnostic decisions are based on specific waveform patterns, which are reflected in reports through textual descriptions. However, existing multi-modal approaches often overlook the alignment between localized waveform features and diagnostic semantics. To the best of our knowledge, no existing method jointly models time-domain signals, frequency-domain features, and clinical reports within a unified framework. To address this gap, we propose a tri-modal framework that enhances both modality diversity and alignment granularity, particularly over MERL Liu et al. (2024), by incorporating spectrogram features and introducing dual-level semantic alignment for more precise cross-modal understanding.

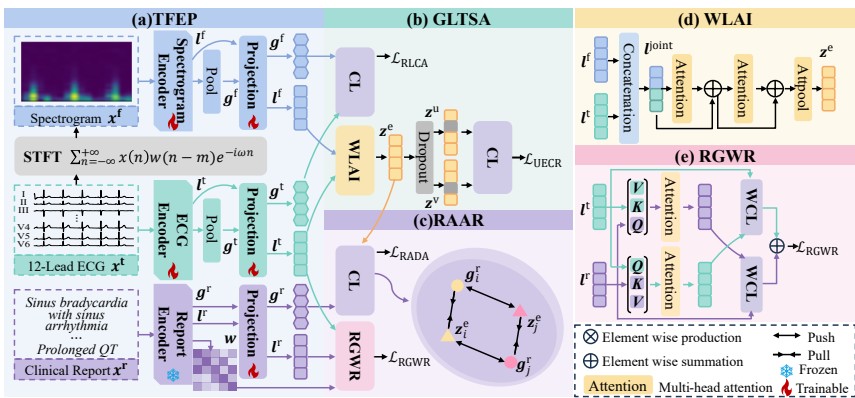

Figure 1: Overview of the TAMER framework. (a) Tri-modal Feature Encoding and Projection (TFEP). (b) Global-Local Temporal-Spectral Alignment (GLTSA). (c) Report-Aware Alignment and Refinement (RAAR). Subfigures (d) and (e) provide detailed illustrations of the internal mechanisms of the Wave-Level Attentive Interaction (WLAI) and Report-Guided Wave-Level Refinement (RGWR) respectively. **CL** and **WCL** denote the contrastive loss and the weighted contrastive loss.

## 3 METHODS

Figure 1 illustrates the proposed TAMER framework. Given an unlabeled tri-modal dataset, $\mathcal{D} = \{(x_i^{\mathrm{t}}, x_i^{\mathrm{f}}, x_i^{\mathrm{r}})\}_{i=1}^N$, each training sample consists of a 12-lead ECG signal $x_i^{\mathrm{t}}$, its corresponding spectrogram $x_i^{\mathrm{f}}$, and the associated clinical report $x_i^{\mathrm{r}}$. The TFEP module encodes each modality to extract both global features $(g_i^{\mathrm{t}}, g_i^{\mathrm{f}}, g_i^{\mathrm{r}})$ and local features $(l_i^{\mathrm{t}}, l_i^{\mathrm{f}}, l_i^{\mathrm{r}})$. Subsequently, the GLTSA performs rhythm-level contrastive alignment on $(g_i^{\mathrm{t}}, g_i^{\mathrm{f}})$ to ensure cross-modal consistency, and wave-level attentive interaction on $(l_i^{\mathrm{t}}, l_i^{\mathrm{f}})$ to produce a unified ECG representation $z_i^{\mathrm{e}}$. $z_i^{\mathrm{e}}$ is further regularized through dropout-based perturbation to produce augmented views for consistency learning. Simultaneously, both $z_i^{\mathrm{e}}$ and $l_i^{\mathrm{t}}$ are fed into the RAAR module, which performs dual-level

contrastive learning with $g_i^r$ and $l_i^r$, promoting deep semantic alignment between ECG signals and clinical reports.

## 3.1 Tri-Modal Feature Encoding and Projection

Multi-modal ECG data comprises time-domain signals that capture rhythm and waveform characteristics, spectrograms that reveal frequency-domain anomalies (e.g., transient bursts, non-stationary spectral patterns), and clinical reports that contain diagnostic knowledge. We hypothesize that modality-specific information is essential for learning high-quality representations.

We propose the tri-modal feature encoding and projection (TFEP) module, as illustrated in Figure 1(a). First, we transform the raw ECG waveform $x_i^t \in \mathbb{R}^{L \times T}$, where $L$ denotes the number of leads (typically 12) and $T$ is the temporal length, into a spectrogram $x_i^f \in \mathbb{R}^{L \times F \times M}$ using the short-time Fourier transform (STFT) Bui et al. (2024). $F$ and $M$ are the numbers of frequency bins and temporal frames, respectively. We then extract local features $l_i^t$ and $l_i^f$ from $x_i^t$ and $x_i^f$ via the hidden layers of the ECG and spectrogram encoders. We also extract global features $g_i^t$ and $g_i^f$ by applying average pooling over $l_i^t$ and $l_i^f$. For the clinical report $x_i^r$, we apply a frozen report encoder to obtain the global representation $g_i^r$ and local representations $l_i^r$, along with the attention weights of the [CLS] token, represented as a vector $w$, which is used to estimate the importance of each report token based on its contribution to the [CLS] token. All extracted features are then projected into modality-specific latent spaces to facilitate downstream contrastive alignment and fusion.

## 3.2 Global-Local Temporal-Spectral Alignment

ECG signals and spectrograms exhibit periodicity and semantic complementarity in the time-frequency domain. To fully leverage this internal consistency, we introduce three modules: (1) rhythm-level contrastive alignment (RLCA) for enforcing global temporal-spectral consistency. (2) wave-level attentive interaction (WLAI) module for enhancing local feature interaction and derive a unified ECG representation, and (3) uni-ECG consistency regularization (UECR) for improving the robustness of the unified representation, as shown in Figure 1(b).

### 3.2.1 Rhythm-Level Contrastive Alignment.

Although temporal and spectral features originate from the same physiological signal, the STFT transformation introduces differences in temporal resolution. Coupled with modality-specific noise, this often lead to semantic misalignment between global cardiac rhythms across modalities, hindering the model's ability to detect periodic abnormalities. To address this, we propose the RLCA, which enforces global alignment between temporal and spectral modalities via contrastive learning. Specifically, inspired by the contrastive learning Radford et al. (2021), given a pair of features $(\boldsymbol{\eta}^a, \boldsymbol{\eta}^b)$ from modalities $a$ and $b$, we minimize the distance between positive pairs $(\boldsymbol{\eta}_i^a, \boldsymbol{\eta}_i^b)$, while maximizing that between negative pairs $(\boldsymbol{\eta}_i^a, \boldsymbol{\eta}_j^b)$. The contrastive loss (CL) $\mathcal{L}_{\text{CL}}(\cdot)$ is defined as:

$$\mathcal{L}_{i,j}^{a2b} = -\log\left(\frac{\exp(\text{sim}(\boldsymbol{\eta}_i^a, \boldsymbol{\eta}_i^b)/\tau)}{\sum_{j=1}^N \mathbf{1}_{[j \neq i]} \exp(\text{sim}(\boldsymbol{\eta}_i^a, \boldsymbol{\eta}_j^b)/\tau)}\right), \quad \mathcal{L}_{i,j}^{b2a} = -\log\left(\frac{\exp(\text{sim}(\boldsymbol{\eta}_i^b, \boldsymbol{\eta}_i^a)/\tau)}{\sum_{j=1}^N \mathbf{1}_{[j \neq i]} \exp(\text{sim}(\boldsymbol{\eta}_i^b, \boldsymbol{\eta}_j^a)/\tau)}\right),$$

$$\mathcal{L}_{\text{CL}}(\boldsymbol{\eta}^a, \boldsymbol{\eta}^b) = \frac{1}{2B} \sum_{i=1}^N \sum_{j=1}^N \left(\mathcal{L}_{i,j}^{a2b} + \mathcal{L}_{i,j}^{b2a}\right), \tag{1}$$

where $\tau$ is the temperature parameter, $B$ is the batch size, $\mathbf{1}(\cdot)$ is the indicator function, and $\text{sim}(\cdot)$ denotes the cosine similarity.

For temporal-spectral pairs $(g^t, g^f)$, the loss $\mathcal{L}_{\text{RLCA}}$ is defined as $\mathcal{L}_{\text{RLCA}} = \mathcal{L}_{\text{CL}}(g^t, g^f)$. By enforcing this global rhythm-level alignment, RLCA enhances the temporal-spectral consistency and improves the model's ability to recognize rhythmic patterns and periodic abnormalities.

### 3.2.2 Wave-Level Attentive Interaction.

While RLCA captures global rhythm consistency, it lacks fine-grained modeling of diagnostic waves, such as the QRS complex or ST segment, which reflect critical pathological patterns across cycles. To this end, we introduce the WLAI module, which enhances wave-level interactions and

constructs a unified ECG representation. Specifically, local temporal and spectral features, $l_i^{\text{t}}$ and $l_i^{\text{f}}$ are concatenated to preserve modality-specific characteristics. A two-stage residual attention mechanism is then applied to adaptively reweigh salient features and align complementary semantics. Finally, a learnable class token and attention-based pooling are incorporated to aggregate diagnostic-sensitive waves into a compact embedding $z_i^{\text{e}}$, which captures compound or co-existing pathological patterns. The overall process is:

$$l_i^{\text{joint}} = \text{concatenation}(l_i^{\text{t}}, l_i^{\text{f}}), \qquad z_i^{(1)} = l_i^{\text{joint}} + \text{att}(l_i^{\text{joint}}), \quad z_i^{(2)} = z_i^{(1)} + \text{att}(z_i^{(1)}), \quad z_i^{\text{e}} = \text{attpool}(z_i^{(2)}), \quad (2)$$

where att denotes the multi-head attention mechanism, and $\text{attpool}(\cdot)$ represents attention-based aggregation (Vaswani et al., 2017).

Unlike conventional fusion methods that directly sum or concatenate modalities, often leading to information redundancy or loss, WLAI selectively integrates clinically relevant features, resulting in a coherent and semantically enriched representation.

### 3.2.3 Uni-ECG Consistency Regularization.

Although the WLAI produces a unified representation, it may still be instability due to modality-specific noise, motion artifacts, or modality discrepancies. To enhance robustness and representation consistency, we introduce the UECR module. UECR aims to enforce view-invariant representations by perturbing the fused embedding $z_i^{\text{e}}$. Specifically, we apply dropout to to generate two stochastic views $z_i^{\text{u}}$ and $z_i^{\text{v}}$. We then apply a contrastive loss, $\mathcal{L}_{\text{UECR}} = \mathcal{L}_{\text{CL}}(z^{\text{u}}, z^{\text{v}})$, using the contrastive function defined in Eq. equation 1 to encourage their alignment in the embedding space.

## 3.3 Report-Aware Alignment and Refinement

In clinical practice, diagnostic reports interpret ECG signals from a medical perspective, providing high-level complementary information. Exploiting the complementarity of these two modalities enhances the model's ability to capture disease-related features. To this end, the RAAR module leverages a frozen text encoder to provide stable diagnostic semantics and enhances ECG representations through two sub-modules: the report-anchored diagnostic-level alignment (RADA) enhances the model's awareness of global diagnostic semantics, while the report-guided wave-level refinement (RGWR) strengthens attention to key diagnostic waves and improves the identification of fine-grained abnormalities, as illustrated in Figure 1(c).

### 3.3.1 Report-Anchored Diagnostic-Level Alignment.

The representation $z_i^{\text{e}}$ produced by the WLAI module integrates key waveform segments to form a comprehensive diagnostic embedding, while the global report embedding $g_i^{\text{r}}$ captures diagnostic semantics from textual descriptions. These two modalities respectively provide the physiological and clinical perspectives necessary for cardiovascular disease diagnosis. To align them, the RADA module enforces global semantic consistency across modalities via contrastive learning. The objective is defined as $\mathcal{L}_{\text{RADA}} = \mathcal{L}_{\text{CL}}(z^{\text{e}}, g^{\text{r}})$ (Eq. equation 1).

### 3.3.2 Report-Guided Wave-Level Refinement.

While RADA captures global semantics, it lacks fine-grained alignment between ECG and diagnostic report. Since WLAI already fuses temporal and spectral features, RGWR focuses on local interactions between temporal ECG waves and diagnostic reports, enabling wave-level semantic refinement and improving abnormality localization.

Let $T_i = \{t_i^k\}_{k=1}^K$, $R_i = \{r_i^m\}_{m=1}^M$, $t_i^k, r_i^m \in \mathbb{R}^D$ denote the local feature sets from ECG recordings and report respectively. A dual cross-attention mechanism computes contextual representations:

$$c_i^k = \sum_{m=1}^M \text{softmax}\left(\frac{Qt_i^k \cdot Kp_i^m}{\sqrt{D}}\right) \cdot Vr_i^m, \quad c_i^m = \sum_{k=1}^K \text{softmax}\left(\frac{Qr_i^m \cdot Kt_i^k}{\sqrt{D}}\right) \cdot Vt_i^k, \quad (3)$$

with $Q, K, V \in \mathbb{R}^{D \times D}$ as learnable matrices.

To dynamically weight the diagnostic importance of different tokens and ECG waves, we utilize token-level attention weights $w_i^j$ generated by the report encoder. The weighted contrastive loss

Table 1: Comparison of different methods on the PTBXL-Super dataset (best in **bold**).

| Method | Training Ratio | Zero-Shot PTBXL-Super | Domain Shift CPSC2018 | CSN |
|---|---|---|---|---|
| SimCLR | 100% | × | 69.62 | 73.05 |
| BYOL | 100% | × | 70.27 | 74.01 |
| BarlowTwins | 100% | × | 68.98 | 72.85 |
| MoCo-v3 | 100% | × | 69.41 | 73.29 |
| SimSiam | 100% | × | 70.06 | 73.92 |
| TS-TCC | 100% | × | 71.32 | 75.16 |
| CLOCS | 100% | × | 68.79 | 72.64 |
| ASTCL | 100% | × | 69.23 | 73.18 |
| CRT | 100% | × | 70.15 | 74.08 |
| ST-MEM | 100% | × | **76.12** | **84.50** |
| MERL | 0% | 74.2 | **88.21** | 78.01 |
| C-MET | 0% | 76.2 | 72.09 | 79.11 |
| **TAMER** | 0% | **76.5** | 84.71 | 80.95 |

(WCL) for local ECG-report refinement denoted as $\mathcal{L}_{\text{RGWR}}$ and computed as follows:

$$\mathcal{L}_{\text{ECG}} = -\frac{1}{2NK} \sum_{i=1}^{N} \sum_{j=1}^{K} \boldsymbol{w}_i^j \log \left( \frac{\exp(\text{sim}(\boldsymbol{t}_i^j, \boldsymbol{c}_i^j)/\lambda)}{\sum_{k=1}^{K} \exp(\text{sim}(\boldsymbol{t}_i^j, \boldsymbol{c}_i^k)/\lambda)} \right), \quad \mathcal{L}_{\text{report}} = -\frac{1}{2NM} \sum_{i=1}^{N} \sum_{j=1}^{M} \boldsymbol{w}_i^j \log \left( \frac{\exp(\text{sim}(\boldsymbol{r}_i^j, \boldsymbol{c}_i^j)/\lambda)}{\sum_{m=1}^{M} \exp(\text{sim}(\boldsymbol{r}_i^j, \boldsymbol{c}_i^m)/\lambda)} \right),$$

$$\mathcal{L}_{\text{RGWR}} = \mathcal{L}_{\text{ECG}} + \mathcal{L}_{\text{report}},$$

$$(4)$$

where $\lambda$ is the temperature parameter. The RGWR highlights waveform segments that are most relevant to the diagnostic report, enhancing interpretability and fine-grained disease recognition. Attention weights $\boldsymbol{w}_i^j$ dynamically adjust focus based on the diagnostic importance of each token, improving sensitivity to critical abnormalities.

By integrating global and local contrastive losses, the RAAR alignment loss is defined as:

$$\mathcal{L}_{\text{RAAR}} = \mathcal{L}_{\text{RADA}} + \mathcal{L}_{\text{RGWR}}. \tag{5}$$

## 3.4 OVERALL LOSS FUNCTION

Finally, TAMER jointly optimizes the key loss functions as follows:

$$\mathcal{L}_{\text{total}} = \mathcal{L}_{\text{RLCA}} + \mathcal{L}_{\text{UECR}} + \mathcal{L}_{\text{RAAR}}. \tag{6}$$

By integrating these components, TAMER significantly enhances cross-modal consistency and improves representation robustness.

## 4 EXPERIMENTS AND RESULTS

### 4.1 PRE-TRAINING

#### 4.1.1 PRE-TRAINING DATASET.

We pre-train our model on the MIMIC-ECG dataset Gow et al. (2023), which contains 800,035 ECG-report pairs collected from 161,252 patients. Each sample consists of a 12-lead ECG signal recorded at 500 Hz for 10 seconds, accompanied by a structured or free-text diagnostic report. Data processing follows the standard pipeline proposed in MERL Liu et al. (2024), including signal normalization, clinical report cleaning, and semantic filtering. After processing, a total of 771,693 high-quality triplets are retained for unsupervised tri-modal training.

#### 4.1.2 PRE-TRAINING IMPLEMENTATION DETAILS.

Our model is implemented in PyTorch and trained on a single NVIDIA A100-PCIE-40GB GPU. The ECG encoder adopts a randomly initialized 1D ResNet-34 He et al. (2016), the spectrogram encoder is a 2D CNN, and the report encoder employs a frozen Med-CPT Query Encoder Jin et al. (2023) for semantic stability.

We use the AdamW optimizer with an initial learning rate of $2 \times 10^{-4}$ and a weight decay of $1 \times 10^{-7}$. To dynamically adjust the learning rate during training, we adopt a cosine annealing warm restart scheduler with an initial restart period of $T_0 = 40,000$. The temperature factor $\lambda$ is set to be 0.04. The model is trained for 50 epochs with a batch size of 256.

Table 2: Comparison of different methods on the CPSC2018 dataset (best in **bold**).

| Method | Training Ratio | Zero-Shot CPSC2018 | Domain Shift PTBXL-Super | CSN |
|---|---|---|---|---|
| SimCLR | 100% | × | 56.65 | 66.36 |
| BYOL | 100% | × | 57.32 | 67.56 |
| BarlowTwins | 100% | × | 55.97 | 65.89 |
| MoCo-v3 | 100% | × | 56.54 | 66.12 |
| SimSiam | 100% | × | 57.21 | 67.48 |
| TS-TCC | 100% | × | 58.47 | 68.34 |
| CLOCS | 100% | × | 55.86 | 65.73 |
| ASTCL | 100% | × | 56.61 | 66.27 |
| CRT | 100% | × | 57.39 | 67.62 |
| ST-MEM | 100% | × | 62.27 | 75.19 |
| MERL | 0% | 82.8 | 76.77 | 76.56 |
| C-MET | 0% | 80.1 | 77.12 | 82.91 |
| **TAMER** | 0% | **88.3** | **82.00** | **86.73** |

Table 3: Comparison of different methods on the CSN dataset (best in **bold**).

| Method | Training Ratio | Zero-Shot CSN | Domain Shift PTBXL-Super | CPSC2018 |
|---|---|---|---|---|
| SimCLR | 100% | × | 59.74 | 62.11 |
| BYOL | 100% | × | 60.39 | 63.24 |
| BarlowTwins | 100% | × | 58.76 | 61.35 |
| MoCo-v3 | 100% | × | 59.82 | 62.07 |
| SimSiam | 100% | × | 60.23 | 63.09 |
| TS-TCC | 100% | × | 61.55 | 64.48 |
| CLOCS | 100% | × | 58.69 | 61.27 |
| ASTCL | 100% | × | 59.74 | 61.12 |
| CRT | 100% | × | 60.48 | 63.33 |
| ST-MEM | 100% | × | 73.05 | 64.66 |
| MERL | 0% | 74.4 | 74.15 | 82.86 |
| C-MET | 0% | 76.3 | 76.24 | 80.10 |
| **TAMER** | 0% | **78.7** | **76.49** | **87.62** |

## 4.2 DOWNSTREAM TASKS

### 4.2.1 DOWNSTREAM TASK DATASETS.

To evaluate the generalization of our pre-trained TAMER across various clinical scenarios, we conduct downstream experiments on three public ECG datasets, each providing 12-lead signals (500 Hz, 10 s). Data splitting and preprocessing follow the protocol in MERL Liu et al. (2024).

**PTBXL-Super Dataset.** A PTBXL subset Wagner et al. (2020) with 21,837 ECG recordings from 18,885 patients across five major CVD categories is used for evaluation.

**CSN Dataset.** The CSN dataset Zheng et al. (2020; 2022) includes 23,026 ECG recordings annotated with 38 diagnostic labels.

**CPSC2018 Dataset.** The CPSC2018 dataset Liu et al. (2018) comprises 6,877 12-lead ECG recordings, with durations ranging from 6 to 60 seconds, and includes 9 diagnostic labels. We retain recordings with durations $\geq$ 10 seconds and truncate all signals to 10 seconds, resulting in 6,867 records used for evaluation.

### 4.2.2 DOWNSTREAM TASK IMPLEMENTATION DETAILS.

To comprehensively assess the generalization ability of the pre-trained model under real-world clinical constraints, we design two zero-shot evaluation scenarios:

**Zero-Shot Classification Across Unseen Labels.** We evaluate the model's ability to recognize previously unseen disease categories while keeping all pre-trained parameters frozen. We adopt the CKEPE prompt dictionary Liu et al. (2024) to generate class-level textual descriptions. The similarity between each ECG representation and the semantic prompt is computed and used as the prediction score on the downstream test set.

**Zero-Shot Classification Under Domain Shift.** To simulate domain shifts frequently encountered in clinical environments, such as varying patient populations or acquisition protocols, we conduct cross-dataset evaluations where the source and target domains share semantically aligned diagnostic labels but differ in data distribution. Label mapping and merging are performed following the pro-

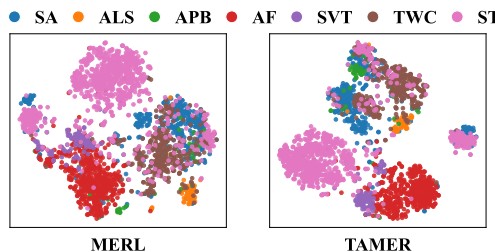

Figure 2: t-SNE Visualization of ECG Features (MERL vs. TAMER) on CSN Dataset

tocol in MERL Liu et al. (2024). The model is directly evaluated on the target domain without any additional tuning.

### 4.3 EVALUATION METRICS

All evaluations are based on macro-AUC, ensuring a fair comparison in the presence of class imbalance and providing a robust measure of model reliability across domains.

### 4.4 COMPARISON WITH STATE-OF-THE-ART METHODS

To comprehensively evaluate the effectiveness and generalization capability of the proposed TAMER framework, we conduct systematic comparisons with state-of-the-art SSL methods under two evaluation settings: zero-shot classification and zero-shot classification under domain shift. Zero-shot classification is performed for multi-modal methods to assess their capacity for semantic understanding and classification. For zero-shot classification under domain shift, multi-modal approaches such as MERL Liu et al. (2024) and C-MET PHAM et al. (2024) are evaluated on the target domain without any fine-tuning, better reflecting their ability to generalize across domains through cross-modal semantic alignment. All uni-modal SSL baselines: SimCLR Chen et al. (2020), BYOL Grill et al. (2020), BarlowTwins Zbontar et al. (2021), MoCo-v3 Chen et al. (2021), Sim-Siam Chen & He (2021), TS-TCC Eldele et al. (2021), CLOCS Kiyasseh et al. (2021), ASTCL Wang et al. (2023), CRT Zhang et al. (2023), ST-MEM Na et al. (2024) are fine-tuned on 100% of the labeled source domain data and then evaluated on the target domain.

#### 4.4.1 ZERO-SHOT CLASSIFICATION.

As illustrated in Tables 1, 2, and 3,TAMER achieves the best overall performance, with AUCs of 76.5%/88.3%/78.7% on PTBXL-Super, CPSC2018, and CSN, respectively. This notably surpasses MERL (74.2%/82.8%/74.4%) and C-MET (76.2%/80.1%/76.3%), especially on CPSC2018 where TAMER an AUC of 88.3%, demonstrating its exceptional capability in handling complex, multi-label ECG classification under zero-shot settings.

#### 4.4.2 ZERO-SHOT CLASSIFICATION UNDER DOMAIN SHIFT.

Under domain shift conditions, TAMER continues to outperform all compared SSL, achieving an average AUC of 81.2% on three cross-domain evaluation settings, without any fine-tuning, as shown in Tables 1, 2, and 3. This highlights its robustness to distribution shifts and the advantage of leveraging tri-modal semantics.

Several insights can be drawn from the results. First, under domain shift scenarios, multi-modal methods generally outperform uni-modal approaches that rely on source-domain fine-tuning, emphasizing the effectiveness of incorporating clinical text as semantic priors to mitigate the impact of distributional shifts. Second, TAMER achieves average AUCs of 81.2% and 83.1% across three datasets on the two downstream tasks, consistently surpassing all uni-modal and multi-modal self-supervised baselines. These results demonstrate its superior generalization capability and validate the effectiveness of our proposed tri-modal architecture and report-aware alignment and refinement module in improving cross-domain modeling.

## 4.5 ANALYSIS OF TAMER

### 4.5.1 ABLATION STUDY.

We conduct ablation experiments by individually removing the RLCA, WLAI, and RGWR modules to evaluate their contributions to model performance. As shown in Table 4, removing RLCA, WLAI, and RGWR results in average AUC drops of 0.57%, 5.03%, and 2.46%, respectively, on the zero-shot task, and 1.59%, 4.14%, and 3.64% on the domain shift task. These results clearly demonstrate the essential role of all three modules in enhancing the model's generalization and diagnostic performance. Specifically, RLCA and WLAI enhance semantic consistency between temporal and spectral modalities at the global and local levels, respectively. RGWR performs fine-grained semantic alignment between ECG signals and diagnostic phrases, facilitating the precise identification of critical abnormalities. Together, these three modules promote deep interaction and semantic enrichment across ECG tri-modal features, significantly improving the model's discriminative capability and clinical adaptability in automated ECG diagnosis. These findings validate the effectiveness of the proposed method in multi-modal medical scenarios.

Table 4: Results of ablation experiments on key modules.

| WLAI | RGWR | Zero-Shot | Domain Shift |
|---|---|---|---|
| × | ✓ | ✓ | 80.62 | 81.49 |
| ✓ | × | ✓ | 76.16 | 78.94 |
| ✓ | ✓ | × | 78.73 | 79.44 |
| ✓ | ✓ | ✓ | **81.19** | **83.08** |

### 4.5.2 EFFECTS OF TEMPERATURE PARAMETER $\lambda$.

The temperature parameter $\lambda$ controls the concentration level of the similarity distributions. We assess the impact of $\lambda$ using values of 0.03, 0.04, and 0.05. As shown in Table 5, $\lambda = 0.04$ yields the best performance in both zero-shot and domain shift settings, indicating a balanced trade-off between training stability and discriminative learning.

Table 5: Effects of temperature parameter $\lambda$.

| $\lambda$ | Zero-Shot | Domain Shift |
|---|---|---|
| 0.03 | 80.42 | 81.66 |
| **0.04** | **81.19** | **83.08** |
| 0.05 | 79.91 | 81.62 |

### 4.5.3 VISUALIZATION OF FEATURE REPRESENTATIONS.

We employ t-SNE to visualize the ECG embeddings extracted from the CSN test set, as shown in Figure 2. To enhance clustering clarity, multi-label samples and classes with fewer than 50 instances are excluded. The visualization shows that TAMER produces more distinct and compact clusters across various diagnostic categories. Compared to MERL, TAMER exhibits clearer group boundaries for easily confusable classes such as ALS, APB, and TWC, indicating improved separation and reduced overlap in the feature space. For well-separated categories like AF and ST, both models perform similarly, confirming that TAMER retains the original discriminative capacity while also producing more refined embeddings for ambiguous cases.

## 5 CONCLUSION

In this work, we propose TAMER, a unified tri-modal pre-training framework for robust and generalizable ECG representation learning. TAMER integrates three key components: a tri-modal feature encoding and projection module, a global-local temporal-spectral alignment module, and a report-aware alignment and refinement module. By jointly modeling ECG signals, spectrograms, and clinical diagnostic reports, TAMER effectively captures heterogeneous and localized semantic information. The fusion and contrastive alignment modules promote consistent, interpretable, and discriminative representations. Extensive evaluations on three public datasets demonstrate that TAMER consistently outperforms state-of-the-art uni-modal and multi-modal baselines in both zero-shot classification and cross-domain transfer tasks.

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

# A    APPENDIX

## A.1    LLM ASSISTANCE STATEMENT

During the preparation of this manuscript, we utilized AI-based assistance tools (OpenAI's Chat-GPT) to support the writing and editing process. The AI was used primarily to:

- Refine and polish the language of certain paragraphs to improve clarity, readability, and conciseness.
- Suggest alternative wording or phrasing for specific terms to enhance precision and academic tone.
- Provide guidance on restructuring sentences or paragraphs for better logical flow.

All scientific content, including experimental design, methodology, results, analysis, and conclusions, was authored and verified solely by the human authors. The AI did not generate any original scientific claims or analyses; it assisted only with language expression and clarity.

## A.2    CHOICES OF TEXT ENCODERS.

We evaluate the effectiveness of different report encoders derived from three representative medical language models: PubMedBERT Gu et al. (2021), Clinical ModernBERT Lee et al. (2025), and Med-CPT Jin et al. (2023). Each text encoder is assessed under identical pre-training and downstream evaluation settings. As shown in Table A1, Med-CPT consistently achieves the best performance across both zero-shot classification and domain shift tasks, significantly outperforming the other encoders. This advantage is attributed to Med-CPT's contrastive pre-training strategy, which is more effective at modeling semantic consistency and capturing fine-grained features in medical reports, thereby improving cross-modal alignment performance.

Table A1: Performance comparison of different text encoders

| Text encoder | Zero-Shot | Domain Shift |
|---|---|---|
| PubMedBERT | 72.23 | 74.36 |
| Clinical ModernBERT | 76.92 | 79.61 |
| **Med-CPT** | **81.19** | **83.08** |

## A.3    COMPARISON OF DIFFERENT METHODS UNDER ZERO-SHOT.

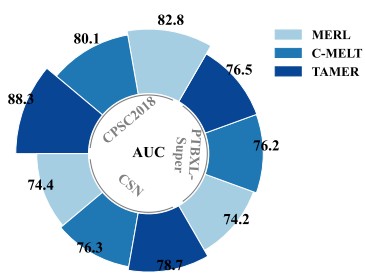

Figure A1: Zero-shot AUC (%) on three ECG datasets: MERL vs. C-MET vs. TAMER. AUC performance (%) of MERL, C-MET, and TAMER across three ECG datasets in the zero-shot setting.

