# OpenReview forum: "TAMER: A Tri-Modal Contrastive Alignment and Multi-Scale Embedding Refinement Framework for Zero-Shot ECG Diagnosis"
_ICLR.cc/2026/Conference — ICLR 2026 Conference Withdrawn Submission_

### Official Review · Reviewer_PPDb · 2025-10-17

**Soundness:** 2
**Presentation:** 3
**Contribution:** 2
**Rating:** 2
**Confidence:** 5

**Summary:**

This paper introduces a new ECG pre-training framework. Compared to prior signal+text frameworks, the authors further added the spectrogram of ECG in a separate branch. Accordingly, they introduced rhythm-level and wave-level embedding alignment between signal and spectrograms, and report-aware alignment between signal and report. The results on zero-shot setting demonstrate the effectiveness of the proposed approach.

**Strengths:**

1. Performance is further improved compared to the baselines in the paper.

**Weaknesses:**

1. The core idea is to integrate time series, spectrogram, and text for ECG pretraining. However, this had been presented long time ago in AudioCLIP [1]. Moreover, the rationale of integrating spectrogram in ECG is not sufficiently justified. Unlike audio that contains substantial frequency information, ECG signal has limited frequencies. For instance, majority of the ECG spectrogram presented in Fig 1 has nearly zero information (dark blue).
[1] Guzhov, A., Raue, F., Hees, J., & Dengel, A. (2022, May). Audioclip: Extending clip to image, text and audio. In ICASSP 2022-2022 IEEE International Conference on Acoustics, Speech and Signal Processing (ICASSP) (pp. 976-980). IEEE.

2. The baselines in the evaluation do not mark the SOTA results. For example, in uni-modal baselines, ECG-FM [2] was the SOTA. For multi-modal baselines, the authors also missed most recent works [3,4]. All of these should be compared.
[2] McKeen, Kaden, Laura Oliva, Sameer Masood, Augustin Toma, Barry Rubin, and Bo Wang. "ECG-FM: An Open Electrocardiogram Foundation Model." arXiv e-prints (2024): arXiv-2408.
[3] Hung, Manh Pham, Aaqib Saeed, and Dong Ma. "Boosting Masked ECG-Text Auto-Encoders as Discriminative Learners." In Forty-second International Conference on Machine Learning.
[4] Wang, Fuying, Jiacheng Xu, and Lequan Yu. "From Token to Rhythm: A Multi-Scale Approach for ECG-Language Pretraining." In Forty-second International Conference on Machine Learning.

3. The ablation is only for the alignment strategies, without evaluating the benefit of adding the spectrogram as the third modality (the core idea of the paper).

4. The evaluation pipeline only shows zero-shot performance. Why not follow the previous works and evaluate the performance with full fine-tuning, zero-shot, and linear probing?

5. Aligning three modalities could have various combinations. Currently, RLCA and WLAI are for signal and spectrogram only, and RGWR is for signal and text. Why selecting these combination? And why other combinations are excluded?

**Questions:**

1. Table 1-3 take lots of space. Why not merge them into a single table?

2. In Line 217-218, what are the two stages for the residual attention mechanism?

3. In Line 219-220 and Line 226-228, the authors claim WLAI selectively integrates clinically relevant features. However, it is unclear how diagnostic-sensitive waves are selected/learned, or why such a process is able to achieve selective integration.

---

### Official Review · Reviewer_aJ7H · 2025-10-30

**Soundness:** 2
**Presentation:** 3
**Contribution:** 2
**Rating:** 2
**Confidence:** 5

**Summary:**

This paper introduces TAMER, a tri-modal framework for zero-shot ECG diagnosis that integrates ECG signals, spectrograms, and diagnostic reports through contrastive and attentive alignment. The model comprises three modules. Evaluations on three public ECG datasets show state-of-the-art AUC scores and strong cross-domain robustness.

**Strengths:**

The paper presents a tri-modal framework (TAMER) that integrates ECG waveforms, spectrograms, and clinical reports for self-supervised ECG analysis. Its modular design, comprising the Tri-modal Feature Encoding and Projection (TFEP), Global-Local Temporal-Spectral Alignment (GLTSA), and Report-Aware Alignment and Refinement (RAAR) modules, is interesting.
The framework’s emphasis on semantic interpretability and robustness to distribution shifts may add clear clinical and research value.

**Weaknesses:**

The clinical interpretability of TAMER remains limited. Though alignment with diagnostic text is claimed, there is little human expert validation to ensure that the model’s attention aligns with genuine clinical reasoning.
Computational complexity will be high. The tri-modal setup, multi-stage attention, and large-scale contrastive learning make it resource-intensive and potentially impractical for real-time or low-resource clinical applications.
The reliance on pre-trained language models (e.g., Med-CPT) will raise concerns about domain bias and adaptability to institutions using different reporting styles.
The explanation of some equations (e.g., loss functions) lacks intuition, and the visualizations (e.g., t-SNE), while illustrative, do not quantify interpretability gains.
TAMER’s evaluation focuses on AUC metrics only. Complementary metrics such as sensitivity or calibration would better demonstrate clinical reliability.

**Questions:**

Could the authors quantify the computational cost and model size compared to prior multimodal approaches like MERL or C-MET?
Have any clinicians evaluated whether the model’s attention maps or refined embeddings correspond to meaningful ECG features?
How does the system handle ECGs without accompanying reports, and can the tri-modal model degrade gracefully to bi- or uni-modal input?

---

### Official Review · Reviewer_4VZ7 · 2025-10-31

**Soundness:** 2
**Presentation:** 2
**Contribution:** 1
**Rating:** 2
**Confidence:** 4

**Summary:**

The paper introduces TAMER, a tri-modal self-supervised framework for zero-shot ECG diagnosis by jointly learning from raw ECG signals, their spectrograms, and associated clinical reports. The main contribution is a hierarchical alignment strategy that first enforces consistency between temporal and spectral ECG representations via a global-local alignment module (GLTSA). Subsequently, it performs another alignment with text reports through a report-aware alignment  (RAAR). Extensive experiments demonstrate that TAMER achieves good performance on zero-shot classification.

**Strengths:**

The originality lies in the tri-modal formulation and, more importantly, the dual-level alignment with clinical reports, which moves beyond simple global matching to a more fine-grained report-guided refinement of waveform representations.

**Weaknesses:**

The overall framework is not novel and prior work has explored self-supervision in frequency domain with raw signals (e.g., using wavelet transforms). I am not sure why using spectrogram is a good idea in this case given such a low sampling rate of ECG signals.  Likewise, the central claim of requiring a tri-modal setup is not rigorously substantiated by the ablation studies. The experiments fail to include a crucial bi-modal (ECG+Report) baseline that uses the proposed RAAR module, making it impossible to disentangle the gains from adding the spectrogram versus the improved alignment strategy. Furthermore, the reliance on static attention weights from a frozen text encoder in the RGWR module is a strong, potentially flawed assumption, as these weights may not be a valid proxy for diagnostic importance. The paper also completely omits any analysis of computational overhead compared to baselines like MERL and most recent paper DBETA [1], which is a critical consideration given the increased complexity of the third modality and its associated encoders and alignment modules.

[1] Pham et al. "Boosting Masked ECG-Text Auto-Encoders as Discriminative Learners." Forty-second International Conference on Machine Learning.

**Questions:**

The RGWR module's reliance on attention weights $w^r$ from a frozen text encoder is questionable. These weights are pre-determined by the text encoder's original objective and are not adapted to the ECG alignment task. Why should we assume these static weights are a meaningful proxy for the diagnostic importance of specific report tokens when aligning with waveform features? Does this not introduce a strong, potentially incorrect, inductive bias?

The ablation study in Table 4 is insufficient to justify the tri-modal design. The primary comparison should be against a bi-modal (ECG+Report) baseline that uses the same RAAR module but omits the spectrogram and GLTSA. Without this experiment, it is impossible to determine if the gains stem from the novel tri-modal fusion or simply from a superior RAAR alignment. Is it not possible that the entire performance improvement over MERL is attributable to RAAR alone, making the spectrogram modality an unnecessary complication?

The domain shift evaluation protocol (Tables 1-3) constitutes an apples-to-oranges comparison. Uni-modal models benefit from full supervised fine-tuning on source domain labels, while multi-modal models are evaluated zero-shot. This confuse architectural superiority with the training paradigm. A fairer comparison would involve also fine-tuning the full multi-modal model on the source domain labels.

The paper completely omits the computational cost, a critical factor for clinical applicability. Adding a third modality with its own encoder and the GLTSA module must introduce significant overhead. Can you provide a detailed comparison of pre-training time, memory consumption, and, crucially, inference latency against the bi-modal MERL baseline, to justify whether the reported AUC gains warrant the increase in architectural complexity?

The CKEPE prompt dictionary is used for zero-shot evaluation. Was this dictionary developed completely independently of the MIMIC-ECG dataset? Please confirm that the selection of prompt phrases was not informed by analyzing the common terminology or structure of the reports used in your pre-training data, as this would constitute subtle form of data leakage into the evaluation protocol.

What is the reason of omitting [1]?

---

### Official Review · Reviewer_Uug3 · 2025-11-01

**Soundness:** 3
**Presentation:** 3
**Contribution:** 2
**Rating:** 2
**Confidence:** 5

**Summary:**

This paper introduces "TAMER," a self-supervised tri-modal learning framework for zero-shot ECG diagnosis, aiming to overcome limitations of existing uni-modal and bi-modal approaches. TAMER jointly models 12-lead ECG signals, their spectrograms, and clinical diagnostic reports. It comprises three key modules: TFEP for feature encoding, GLTSA for global-local temporal-spectral alignment, and RAAR for report-aware alignment and refinement, which collectively enrich ECG representations with multi-scale and cross-modal semantics. The authors report that TAMER achieves state-of-the-art performance in zero-shot classification and cross-domain generalization across three public ECG datasets.

**Strengths:**

*   **New Tri-modal Approach and Performance Gains:** The attempt to integrate three distinct information sources—ECG time-series, spectrograms, and clinical reports—within a self-supervised learning framework is a novel approach that differentiates it from prior work. The demonstrated superior performance in zero-shot and cross-domain settings, outperforming existing SOTA models, is highly encouraging. This suggests that the complementary information from the three modalities can significantly contribute to learning robust ECG representations.

**Weaknesses:**

1. **Lack of Direct Evidence for "Localized Wave Feature and Semantic Diagnosis Alignment":**
*   The paper states that addressing "local correspondences between waveform anomalies and diagnostic phrases" is a key objective, with the RGWR module in RAAR being responsible for this. However, while the ablation study shows RGWR contributes to overall performance, there is **no direct visual or qualitative evidence** (e.g., attention map visualizations, highlighting of specific waveform anomalies matched with corresponding report phrases) presented in the experiments to convincingly demonstrate *how* this local alignment between ECG wave features and specific diagnostic semantics is achieved or how accurately it performs. This diminishes the persuasiveness of this core claim.

2. **Lack of Distinctiveness and Novelty in Modules:**
*   Many modules are proposed (TFEP, GLTSA with RLCA, WLAI, UECR; RAAR with RADA, RGWR), but the paper struggles to clearly articulate the **fundamental conceptual originality or technical innovation** of each module beyond simply "adding" another component. For instance, it's not clear what makes WLAI's "two-stage residual attention mechanism" or RGWR's "dual cross-attention" truly unique or specifically tailored to the ECG-report domain, or what advantages they offer over standard attention mechanisms. The core contributions of these modules need to be more explicitly highlighted.

3.   **Fundamental Question Regarding the Definition of "Tri-modal":**
*  The spectrogram is generated **deterministically** from the ECG time-series data using a Short-Time Fourier Transform (STFT). It can be argued that the spectrogram is merely another perspective or representation of the same underlying ECG signal, rather than an **independent modality** or a distinct source of information, unlike a patient's clinical metadata (e.g., blood pressure, lab results) or independent medical imaging (e.g., echocardiogram images). From this perspective, claiming "tri-modal" might be an overstatement or misleading.

4.   **Excessive and Non-intuitive Acronyms with Insufficient Explanation:**
*   The paper uses a high number of acronyms (TFEP, GLTSA, RLCA, WLAI, UECR, RAAR, RADA, RGWR) throughout. The meaning of each acronym is not immediately intuitive in the flow of the text, and coupled with the complexity of Figure 1, this makes it difficult for the reader to grasp the entire framework quickly. More clear and intuitive explanations for each module upon its introduction, or a reduction in the overall number of acronyms, would greatly improve readability and reduce cognitive load for the reader.

**Questions:**

1.  **Request for Localized Alignment Evidence:**
*   Please provide **qualitative analysis** to directly demonstrate how the Report-Guided Wave-Level Refinement (RGWR) module within RAAR achieves local semantic alignment between fine-grained ECG waveform features and specific diagnostic phrases from clinical reports. For example, present attention map visualizations or matching examples of particular ECG waveform abnormalities (e.g., QRS complex changes, ST segment elevation) with corresponding text in diagnostic reports (e.g., "ST elevation") to substantiate this claim.

2.  **Clarification of Module Originality and Theoretical Contributions:**
*   Please more clearly explain **what novel ideas, unique adaptations, or theoretical contributions** the individual modules within GLTSA and RAAR (e.g., WLAI, RGWR) offer, beyond merely combining existing deep learning components (attention, contrastive learning). Emphasize the **key differentiators and specific advantages** of these modules in addressing the unique characteristics of the ECG domain and the goal of tri-modal integration.

3.  **Reconsideration or Justification of "Multi-modal" Definition:**
*   Please **reconsider the conceptual validity** of treating ECG time-series signals and their deterministically generated spectrograms as two independent "modalities," or provide a **stronger justification** for this claim. Explain whether spectrograms genuinely offer unique information (e.g., specific abnormal frequency band patterns) that is not directly evident in the time-series, effectively acting as an independent physiological signal. Otherwise, the "tri-modal" claim should perhaps be more accurately reframed, for example, as "bi-modal (ECG-report) with enhanced time-frequency feature representation."

4.  **Improvement of Acronym Usage:**
*   To enhance reader comprehension, please **reduce the number of acronyms** used throughout the paper or ensure that each acronym is thoroughly explained upon its first appearance. Additionally, consider refining diagrams like Figure 1 to make the functional roles of each module more intuitive and immediately understandable.

---

### Note · Authors · 2025-11-12

I have read and agree with the venue's withdrawal policy on behalf of myself and my co-authors.